# Real-time 3D analysis during electron tomography using tomviz

Jonathan Schwartz [1], Chris Harris[2], Jacob Pietryga[1], Huihuo Zheng[3], Prashant Kumar [4], Anastasiia Visheratina[4], Nicholas A. Kotov [4], Brianna Major[2], Patrick Avery [2], Peter Ercius [5], Utkarsh Ayachit[2], Berk Geveci[2], David A. Muller [6], Alessandro Genova[2], Yi Jiang [7], Marcus Hanwell[2,8] & Robert Hovden [1,9] ✉

The demand for high-throughput electron tomography is rapidly increasing in biological and material sciences. However, this 3D imaging technique is computationally bottlenecked by alignment and reconstruction which runs from hours to days. We demonstrate real-time tomography with dynamic 3D tomographic visualization to enable rapid interpretation of specimen structure immediately as data is collected on an electron microscope. Using geometrically complex chiral nanoparticles, we show volumetric interpretation can begin in less than 10 minutes and a high-quality tomogram is available within 30 minutes. Real-time tomography is integrated into tomviz, an open-source and cross-platform 3D data analysis tool that contains intuitive graphical user interfaces (GUI), to enable any scientist to characterize biological and material structure in 3D.

Three-dimensional (3D) characterization across the nanoscale is now possible using scanning/transmission electron microscopes (S/TEM)[1–5]. In an electron tomography experiment, the volumetric structure of biological or material specimens is reconstructed from high-resolution projection images acquired across many viewing angles[6,7]. Unfortunately, tomographic reconstructions can take one to several days to complete depending upon the dataset size or algorithm(s) employed. Even worse, the reconstruction occurs offline, long after all the data has been collected, preventing immediate interpretation during an ongoing experiment. While advancements in detector hardware have boosted throughput with digital data collection[8], substantial human effort and computational resources are still required to process the raw data before visualization. It has been a longstanding goal to begin 3D analysis of specimens in real time to allow immediate assessment of nanoscale structure and data quality[9].

Here we present facile 3D visualization of specimens during an electron or cryo-electron tomography experiment using the tomviz platform (tomviz.org). Our platform now provides interactive 3D material or biological structure in real-time to enhance high-throughput specimen interpretation. Tomviz offers multiple real-time reconstruction algorithms integrated into a fully graphical interface that presents the user with immediate visualization during data collection. Achieving high-throughput electron tomography requires an integrated pipeline that links the microscope hardware to optimized reconstruction algorithms and efficient 3D volumetric visualization. A multi-threaded data analysis pipeline runs dynamic visualizations that update as new data is collected or reconstruction algorithms proceed. Iterative reconstruction algorithms efficiently accommodate new data and keep pace with typical experimental acquisition rates. Scientists can interactively analyze 3D specimen structure concurrent with a tomographic reconstruction after or during an experiment. The robust graphical interface allows for 3D specimens to be rendered as shaded contours or translucent volumes that can be rotated, cropped, or sliced

[1]Department of Material Science and Engineering, University of Michigan, Ann Arbor, MI, USA. [2]Kitware Inc, Clifton Park, NY, USA. [3]Argonne Leadership Computing Facility, Argonne National Laboratory, Lemont, IL, USA. [4]Department of Chemical Engineering, University of Michigan, Ann Arbor, MI, USA. [5]The Molecular Foundry, Lawrence Berkeley National Laboratory, Berkeley, CA, USA. [6]School of Applied and Engineering Physics, Cornell University, Ithaca, NY, USA. [7]Advanced Photon Source, Argonne National Laboratory, Lemont, IL, USA. [8]National Synchrotron Light Source II, Brookhaven National Laboratory, Upton, NY, USA. [9]Applied Physics Program, University of Michigan, Ann Arbor, MI, USA. ✉e-mail: hovden@umich.edu

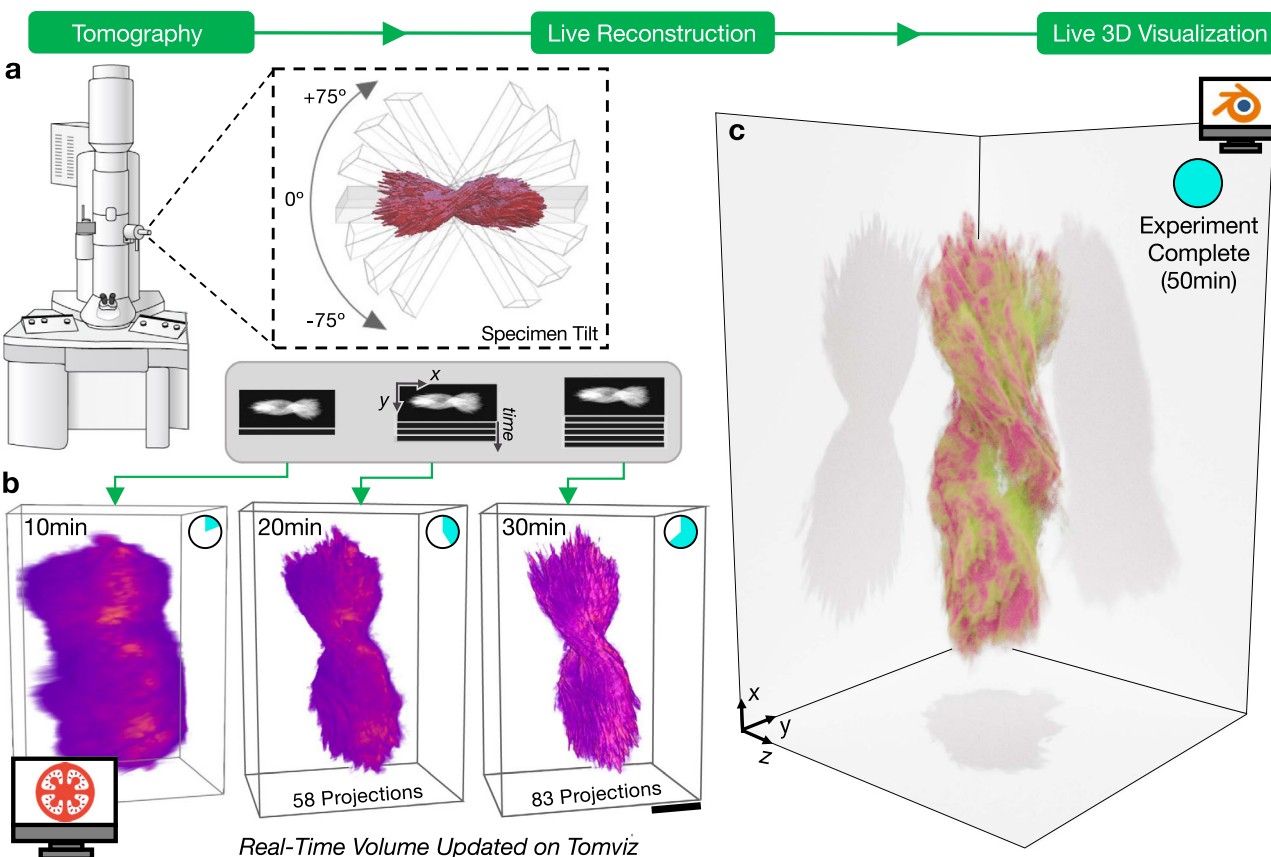

**Fig. 1 | Real-time electron tomography workflow of a helical nanoparticle visualized on tomviz. a** Specimen projections are sequentially collected in an electron microscope across an angular range (<±75°) and continually passed to tomviz for reconstruction and live 3D visualization. **b** As projections accumulate during the experiment, the reconstruction updates in real-time and resolution improves. Scale bar, 100 nm **c** A high-quality tomogram is available for data interpretation upon the end of an experiment.

as the reconstruction occurs. In favorable cases, structural interpretation can begin as early as 10 min and a high-resolution volume is available after only 60% of data is acquired (~30 min). The latest tomviz release (v 2.0) is now packaged with real-time 3D analysis for electron tomography and is available as an open-source cross-platform tool with compiled binaries certified for Linux, Mac, and Windows.

## Results

### Real-time tomography workflow

The real-time tomography workflow is illustrated in Fig. 1: electron micrographs are collected, passed to tomviz for reconstruction, and visualized as an interactive 3D rendering. This process runs simultaneously and continuously while the electron microscope is being operated. During experimental acquisition, tomviz monitors when new projections are collected (Fig. 1a) and appends new data into the reconstruction process. Importantly, tomograms are reconstructed in parallel with data acquisition. Real-time algorithms accommodate the arrival of new data without restarting the reconstruction process. Iterative reconstruction methods are made efficient for real-time processes by utilizing dynamic descent parameters (see "Methods"). Dynamic reconstructions maintain pace with typical experimental acquisitions (e.g., $512^3$–$1024^3$ voxels) using a personal computer. The intermediate reconstructions are rendered in 3D and immediately presented to the scientist (Fig. 1b). Thus, the tomogram dynamically improves with time as both the reconstruction algorithm converges and additional specimen information arrives. High-quality 3D reconstructions are available before the end of the experiment (Fig. 1c).

### Early insight into 3D structure

Direct visualization of a specimen's 3D structure enables immediate identification of morphological and internal information shortly after a tomography experiment begins. We demonstrate real-time tomography on a helical nanoparticle comprised of a chiral dipeptide Cystine amino acid coordinated with Cadmium (Cyst/Cd) (Supplementary Videos S1, S2, and S7). The bowtie-shaped particles were synthesized using a size-limited self-assembly process described by Yan et al.[10]. These semiconducting nanoparticles contain strong tunable chiroptical properties due to a twisted geometry[10]. As shown in Fig. 1b, the overall morphology for the Cd/Cyst nanoparticle can be observed as early as 10 min and fine details are visible after 20–30 min of the experiment (roughly half-completion). The specimen's right-handed chirality cannot be determined from a single projection image and requires 3D imaging (Fig. 1c). With real-time tomography the material's chirality and symmetry were identified within the first third of data acquisition (~15 min). This immediate feedback can save researchers days of effort as reconstructions are no longer processed offline. Moreover, real-time visualization allows quick adjustment and optimization of reconstruction parameters that can greatly influence the reconstruction quality. Ultimately, scientists can efficiently investigate 3D nanostructure during imaging to guide experiments and redefine scientific objectives while simultaneously operating the microscope.

### Real-time 3D visualization during reconstruction

Currently, the best tomographic reconstructions are obtained from algorithms that are slow and iterative. In practice, electron tomography experiments are limited by a finite and restricted angular range

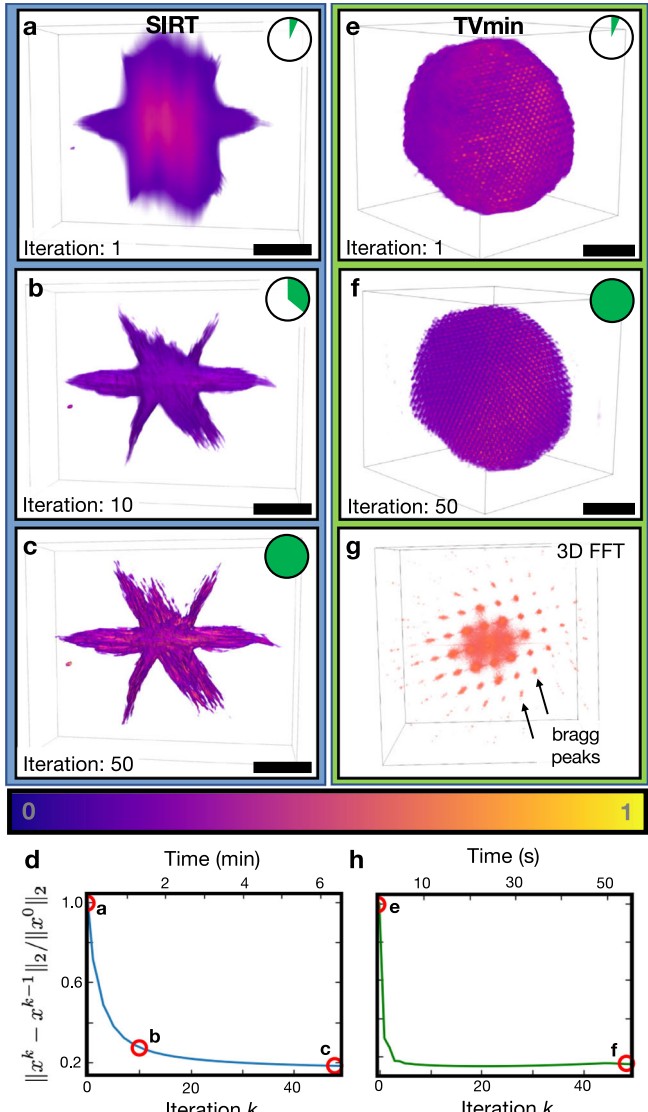

**Fig. 2 | Demonstration of iterative reconstruction algorithms. a–c** Visualization of the $Co_2P$ nanoparticle early, mid, and at the end of the reconstruction process. At the beginning, the underlying structure can partially be seen behind the excess of background intensity. In the middle of the process, sharp features begin to form. The final iteration converges to a tomogram visually similar to the input tilt series. Scale bar, 50 nm. **e–g** Visualization of an atomic resolution FePt nanoparticle. The atoms in the TV nanoparticle are resolved with increasing iteration and its periodicity demonstrated with the fast Fourier transform (FFT). Scale bar, 1 nm. **d**, **h** A plot of the normalized residual to demonstrate convergence.

(e.g., <±70°) resulting in incomplete information that degrades resolution in 3D[11]. Iterative algorithms can recover tomograms with high spatial resolution and minimal reconstruction error[12]. While these algorithms better estimate 3D structure from under-determined measurements, they come at the expense of computational time[13]. Fortunately, using the tomviz tool, iterative reconstructions can be visualized in real-time throughout the arduous computation (Supplementary Videos S2 and S3).

Real-time tomography greatly alleviates the wait-time by visualizing the intermediate 3D structure between algorithm iterations—beneficial during an experiment or analysis. Figure 2 demonstrates interactive visualizations of the Simultaneous Iterative Reconstruction Technique (SIRT)[14] for a cobalt phosphide ($Co_2P$) hyperbranched nanoparticle[15] ($512^3$ voxels volume reconstructed across the 363.52 nm full field of view). SIRT tomograms begin with a loose estimate[16]

(Fig. 2a) and develop sharper, high-frequency information with each increasing iteration (Fig. 2b, c and Supplementary Video S4). Compressed sensing algorithms such as total variation minimization (TVmin) seek maximally sparse solutions to recover high-resolution, low-noise structures using fewer projections than conventional methods[17,18]. Figure 2e, f and Supplementary Video S5 demonstrates an interactive 3D visualization using TVmin reconstruction of an iron platinum (FePt) nanoparticle at atomic resolution ($256^3$ voxels volume reconstructed across the 9.536 nm full field of view)—data provided and pre-processed by Yang et al.[19]. This work replicates the atomic resolution tomogram using independent pre-processing and reconstruction methods. Recent developments in dynamic compressive sensing[20] have also been incorporated into tomviz to accommodate the arrival of new projections during an experiment.

In addition to early estimates of specimen structure, real-time tomography allows assessment of the reconstruction convergence. This is observed qualitatively in the 3D visualization (Fig. 2e, f) and quantitatively plotted in the residuals (Fig. 2d, h). Watching the convergence provides visual inspection and intuition of how hyperparameters influence the final 3D structure and ensure proper convergence. For example, compressed-sensing-inspired reconstruction methods are sensitive to regularization weights and require visual inspection to assess accuracy[21]. Furthermore, these advanced reconstruction algorithms do not exhibit predictable or monotonic convergence a priori and require monitoring to optimize convergence and determine when to terminate[22,23]. Even for traditional algorithms where convergence is more predictable, they are often slow and changes become marginal—the scientist need not wait to begin interpreting the 3D structure. Lastly, practical issues such as misalignment, spurious values in data (e.g., hot pixels), in-plane rotations, and other pre-processing artifacts alter or degrade a reconstruction; however, these problems are diagnosable without completing a full reconstruction. Real-time assessment saves researchers time by providing early feedback and optimizing reconstruction parameters to serve the longstanding goal of high-throughput tomography.

Alternatively, weighted back projection (WBP) reconstructions are ideal for quick assessment of specimen morphology due to their fast, non-iterative computation[24,25]. Figure 3 shows screenshots taken from a live WBP reconstruction visualized using tomviz—time proceeds from left to right. Figure 3a is a tomogram of gold (Au) nanoparticles on strontium titanate (STO) nanocubes. Figure 3b shows platinum (Pt) nanoparticles on a carbon (C) support with the rotation axis along the x-direction. For WBP of single-axis tomography, partial volumetric updates are provided slice by slice along the direction parallel to the rotation axis. In the software, the 3D visualizations dynamically grow in one direction throughout the computation. Supplementary Video S6 shows the user experience of a WBP reconstruction emerging over just a few minutes.

## The live tomography software

The latest tomviz release (v. 2.0) includes real-time tomography capabilities, is entirely open-source (BSD License), runs on all operating systems (OSX, Windows, Linux) with certified installers, and can be implemented on rudimentary TEMs available at most institutions. A user manual with step-by-step instructions for implementing real-time tomography is provided as a Supplementary Protocol along with Supplemental Video demonstrations (Supplemental Videos S3 and S4).

The tomviz graphical user interface (GUI) (Fig. 4) provides an intuitive tomography tool that allows scientists to focus on 3D specimen interpretation[26]. Tomviz monitors data directories for the arrival of new projections during an experiment (Fig. 4a) and visualizes the 3D reconstruction as it dynamically updates (Fig. 4c). During a real-time tomographic reconstruction users can zoom, rotate, slice, and segment the object to highlight regions of interest as the algorithm runs independently. Each voxel in the 3D render (i.e., volumes or isometric

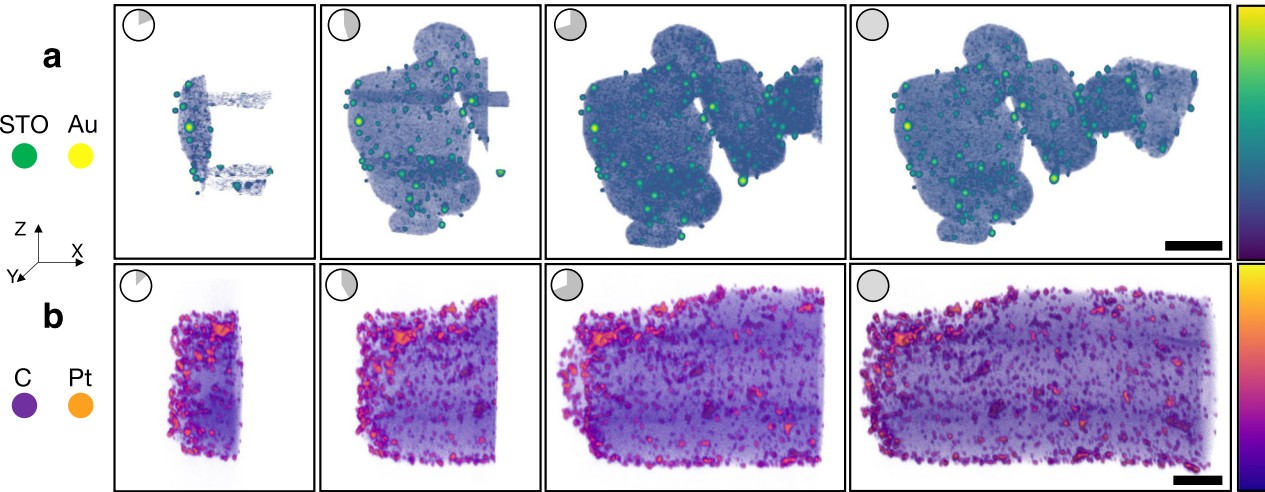

**Fig. 3 | Demonstration of live WBP.** Live tomographic reconstruction in tomviz shown through freeze frames during the progression of a weighted back-projection algorithm (left to right). This unique capability allows users to interact and analyze the 3D structure throughout reconstruction. In the actual software the reconstruction updates in real time. **a** Live volume rendering of Au/strontium titanate (STO) nanocubes. **b** Live volume rendering of platinum (Pt) nanoparticles on a carbon support. Scale bar, 50 nm.

contours) is assigned a color and opacity controlled by the color-opacity transfer function overlaid on the histogram visible at the top of the GUI (Fig. 4, top-right). Users can intuitively define voxel transparency by selecting points on the curve and dragging it between transparent and opaque. The data "Pipeline" retains all transformations and modules performed to produce visualizations, all of which can be saved in a state file for sharing and reproducibility.

A seamless user experience is enabled by an underlying multi-threaded framework of Python/C++ interactions. As the reconstruction occurs, algorithms written in Python trigger signals to notify the application that a new volume is available ("Methods"). Tomographic reconstructions can either run on basic computer infrastructure found on any laptop or scaled across multiple GPUs to process large volumes (>$1024^3$ voxels). Live reconstructions without performance degradation require a tripling of memory requirements. One data copy resides on the visualization (GUI) thread, another on the reconstruction (Python) thread, plus a temporary copy for efficient staging and handoff. The temporary copy allows the reconstruction to run unhindered during the handoff process. The total memory usage for real-time reconstruction is usually well within a consumer-grade computer (c.a. 0.4–16 GB). After the reconstruction is complete, all copies are released from memory and only the final reconstruction remains. Analytical reconstruction methods such as WBP can run slice-by-slice with new reconstructed slices appended along a single reconstruction direction. For iterative methods, we recommend updating the entire volume either every iteration or every few (depending on the speed of computation)—especially for complex sampling schemes such as dual or multiple-axis tomography that lacks a single rotation axes. Enhancements to the underlying 3D rendering (VTK) within tomviz were made to improve interactive visualization and analysis throughout the reconstruction process[27].

## Discussion

We demonstrate real-time visualization of electron tomography reconstructions as they proceed during or after an experiment using tomviz, an open-source cross-platform tool compatible with all electron microscopes. We achieved real-time electron tomography by integrating dynamic volumetric data analysis tools, data input/output, processing, reconstruction, and visualization into a single software tool. In the actual software, the 3D visualizations are dynamically updated in parallel with computation. This means that scientists need not wait for reconstruction to complete, or all data to be collected before beginning the interpretation of results. Continuous feedback provides high-throughput and early diagnoses of 3D specimens, opportunities to optimize experimental parameters, or investigate multiple regions of interest. Although dose is fundamentally set by the experimental acquisition parameters (e.g., dwell time, beam current, sampling rate, or tilt increment), in practice real-time tomography may reduce dose by streamlining acquisition and allowing the possibility of early termination if the reconstruction requirements are met. Optimized, threaded pipelines and the iterative nature of tomographic methods allow tomviz to show intermediate results with minimal impact on performance. This enables interactive 3D analysis of the current reconstruction state while the reconstruction proceeds on a separate thread. A robust graphical interface allows objects to be rendered as shaded contours or volumetric projections and these objects can be rotated, cropped, or sliced. This capability opens radically new possibilities for developing high-throughput, real-time tomographic reconstruction algorithms for geometrically complex inorganic[28] or biological materials. Ultimately, interactive real-time visualization goes beyond high-throughput and allows researchers to make early judgments to answer or identify new scientific questions during experimentation.

## Methods
### Installing tomviz
Tomviz binary installers are available at tomviz.org for macOS, Linux, or Windows[29]. A user manual for real-time tomography using tomviz is provided as Supplemental Material.

### Source-code availability
In addition to stable binary releases, the latest experimental builds and source code is available at github.com/OpenChemistry/tomviz. The entire package is built from 74,029 lines of code and 5253 merges to date[30]. The application is primarily developed in C++ using CMake to coordinate the build process. Algorithms for electron tomography are primarily written using Python to offer facile in-app readability and modification. The entire code-base and dependencies are open-source and compiled to maximize reproducibility.

### License
Tomviz and its underlying tomography algorithms are developed openly and freely as open source software under the 3-clause BSD License[31]—an Open Source Initiative approved license. This allows for

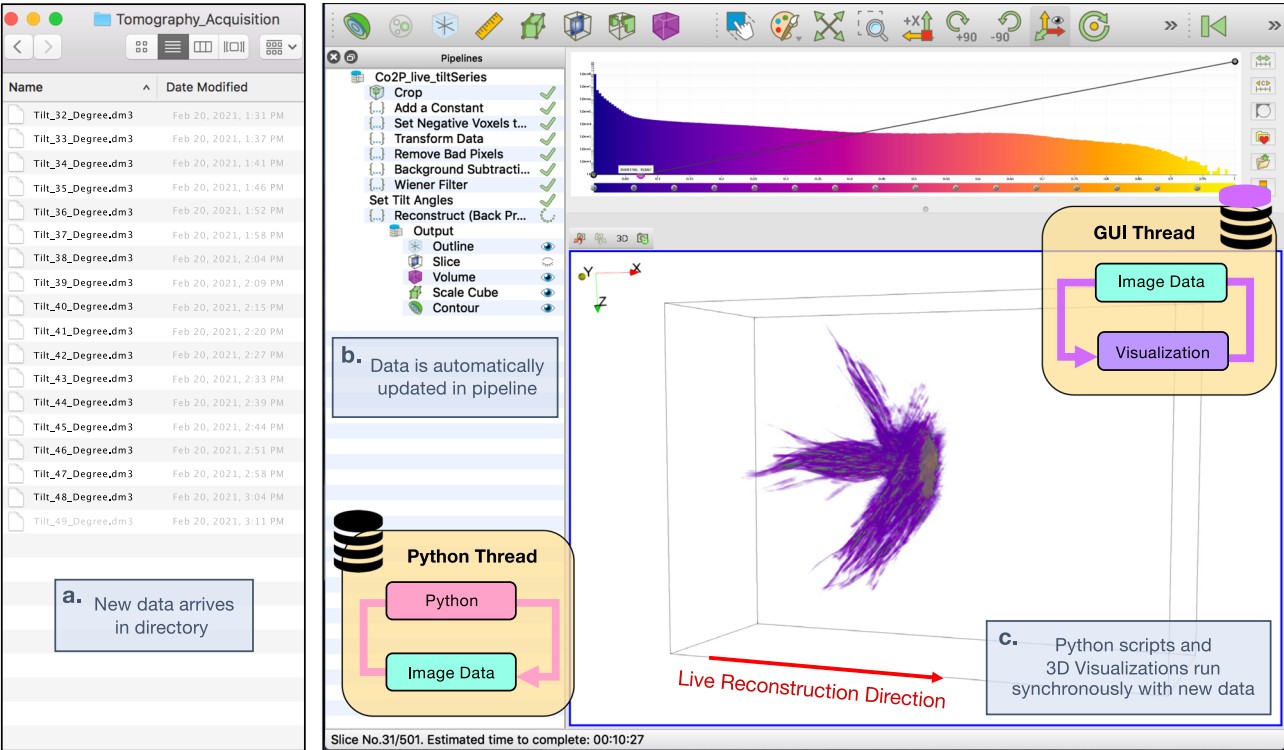

**Fig. 4 | External and internal architecture of tomviz GUI.** The tomviz platform is composed of a multi-threaded pipeline that synchronously handles tomographic and 3D visualization on separate threads. **a** Tomviz monitors for recently acquired tilt projections within a directory and **b** automatically reads new data into the pipeline. **c** As tomographic reconstructions proceed, visualizations dynamically update and remain interactive for analysis.

unrestricted academic, commercial, and government use with no obligation on the part of the licensee to distribute the source code. This license encourages the widest possible re-use of the source code.

### Specimen synthesis and preparation

Cyst/Cd helical nanoparticles are self-assembled via electrostatic and coordination interactions between positively charged cadmium ions and negatively charged chiral dipeptide cystine. The helical nanoparticles were mixed in an aqueous solution and drop cast using a micropipet onto a 3 mm copper TEM grid dried at room temperature. The TEM grid was an ultrathin (3 nm) carbon film with a large hexagonal mesh (100) to provide high specimen tilts without beam shadowing (Electron Microscopy Sciences, Hatfield, PA, USA). Specimen preparation for the $Co_2P$[32], C/Pt[32], FePt,[19] and Au/STO[33] datasets are available in the cited manuscripts with data descriptors.

### Electron tomography acquisition

Real-time electron tomography of the Cys/Cd helical nanoparticle (Fig. 1) was performed during experimental image acquisition on a Talos F200X (Thermo Fisher) operated at 200 kV with a 10.5 mrad semi-convergence angle using an annular dark field detector with an inner collection angle of 36 mrad. The projections were recorded from −64° to +71° with a +1° angular increment using a Model 2021 Fischione Analytical Tomography Holder. At each tilt angle, a STEM image with a 4 µs dwell time at each pixel of lateral dimension 2.47 nm. The tilt series for Figs. 2–4 were collected and aligned in advance of the real-time reconstruction. The FePt nanoparticle (Fig. 2f) was collected on a FEI Titan at 300 keV with a 30 mrad convergence angle and 1.5° tilt increment[19]. The $Co_2P$ (Fig. 2c), Au/STO (Fig. 3a), and C/Pt (Fig. 3b)[32] nanoparticles[33] were acquired on a FEI Tecnai F20 at 200 keV and 2°, 2°, and 1° tilt intervals, respectively. Additional experimental information is available in the corresponding references for each dataset.

A user manual with step-by-step instructions for implementing real-time tomography is provided as a Supplementary Protocol along with supplemental video demonstrations. As with any tomography experiment, microscope alignment is critical. In particular, the sample should be eucentric to alleviate specimen drift and the need for any stage refinement during acquisition. After the microscope is aligned, a user defines the data directory tomviz will monitor to bridge the pipeline from data acquisition to the 3D reconstruction and visualization. Because tomviz operates independently from the microscope acquisition control, this real-time tomography tool can run on any TEM and users can choose their preferred acquisition programs (e.g., Nion Swift[34], Digital Micrograph, FEI Velox, SerialEM[35]).

### File formats

Data stored as raw binaries, XDMF, HDF5, text, png, SER, DM3/4, or TIFF can be read into tomviz. This includes 32-bit IEEE floats. Users can save visualizations and computations as state files (.tvsm) to reproduce results and be shared among colleagues. Reconstructions can be exported into file formats compatible with dedicated 3D rendering software (e.g., Blender).

### Data processing

Tomographic experiments require identification of the center of rotation in the projection tilt series; otherwise, artifacts will be introduced into the tomogram[36]. Even after aligning the stage to eucentricity, the rotation axis can be offset from center and often require additional processing. When an object is tilted around the rotation axis, the object's center of mass (CoM) forms a circle and coincides with the origin of the perpendicular axis. To determine the CoM, we projected each projection onto the perpendicular axis and calculated its shift: $x_{CM} = \sum_i x_i \rho(x_i) / \sum_i \rho(x_i)$ where $\rho(x_i)$ is the Coulomb potential at position $x_i$[4]. This method is known to be sensitive to noise, so prior to aligning the projections we performed a background subtraction to account for the sample support (lacey carbon) and increasing

thickness from high tilts. In our experiments, the uniform thin carbon support was removed by subtracting the average background scattered intensity.

Successful tilt axis identification with the center of mass alignment requires the total projected volume to be fixed for each projection[37] and objects to be isolated. In cases where either of these requirements are not met (e.g., slab specimen geometries or fields of view where multiple particles are visible), alternative alignment routines should be considered. We provide cross-correlation[38] for aligning non-isolated objects (Supplementary Fig. S3). For those that would like to perform marker-based alignment, we suggest using IMOD[39] prior to loading input projections in tomviz. Further tilt axis refinement can be selected with our automated identification script. More advanced iterative projection matching alignment routines can be utilized near the end of data collection to improve the tomogram resolution[40].

### Real-time reconstruction algorithms during experimental acquisition

Modifications to the common implementation for SIRT and TVmin were made to account for the dynamic addition of input projections throughout an experiment. SIRT seeks the minimal error between the reconstruction and experimental data: $\arg\min_x \| Ax - b \|_2$ where $A$ is the measurement matrix, $b$ is the experimental projections, and $x$ is the tomogram. We can further regularize the process through the assumption that our volumes should be piece-wise smooth and minimize its total variation $\|x\|_{TV}$. Iterative algorithms require rescaling of the descent parameter based on the number of projections sampled. SIRT can easily estimate the descent parameter through the calculation of the Lipschitz constant ($L = \|A^T A\|_2$). The Lispchitz constant can be estimated by using the power method[41]. The descent parameter for TVmin is scaled by a dampening envelope that ensures its magnitude decays linearly[20]. Non-iterative algorithms such as WBP do not require rescaling of descent parameters and simply need to reinitialize the computation with the new projection images collected.

### GUI parallelization for real-time visualization

The multithreaded pipeline within the tomviz application executes long-running jobs while simultaneously offering real-time visualization of the progress. As the reconstruction occurs, algorithms written in Python can trigger signals to notify the application that a new volume is available. A slot on the C++ side listens for this signal, using a mutually exclusive lock (mutex) on the image data to secure access to the updated volume. The new data is copied into the foreground thread (main GUI), and once it is available the mutex is released. Once the application receives a signal indicating that the output has been updated, downstream data operations can then be re-executed and any connected visualization modules will also be notified. As an effect, the histogram is recalculated in another background thread while all the current visualization modules display the rendered representation. In the case of the contour module, this will necessitate the recalculation of the surface mesh or the update will be uploaded to the GPU for volume rendering.

### Data availability

Figure 1 and the user manual use exemplary data from Supplemental Dataset 1. The aligned and raw tilt series for the FePt dataset in Fig. 2 can be accessed through physics.ucla.edu/research/imaging/FePt. In addition, the projection images for the $Co_2P$ and C-Pt nanoparticles displayed in Figs. 2 and 3 are available through doi.org/10.6084[32].

### Code availability

Tomviz binaries are available as Supplementary Software and the source code can be downloaded from GitHub (github.com/OpenChemistry/tomviz).

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

## Acknowledgements

R.H. and J.S. acknowledge support from the Army Research Office, Computing Sciences (W911NF-17-S-0002). The tomviz project began with support under DOE Office of Science contract DE-SC0011385. This research used resources of the Argonne Leadership Computing Facility, which is a DOE Office of Science User Facility supported under Contract DE-AC02-06CH11357. P.E. is supported by the Molecular Foundry, Lawrence Berkeley National Laboratory, which is supported by the U.S. Department of Energy under contract no. DE-AC02-05CH11231. The authors thank Suk Hyun Sung and Yanqi Luo for input and feedback. Microscope illustration in Fig. 1 is released under creative commons license by Database Center for Life Science (DBCLS). This work made use of the Michigan Center for Materials Characterization (MC2) with technical support from Tao Ma and Bobby Kerns. The authors acknowledge financial support from the University of Michigan College of Engineering.

## Author contributions

M.H. and R.H. conceived the idea. R.H., M.H., C.H., J.S., D.A.M., P.E., P.A., B.M., B.G., U.A., and A.G. designed tomviz's software and visualization architecture. J.S., J.P., Y.J., H.Z., and R.H. implemented real-time tomography algorithms. J.S., J.P., and R.H. conducted experiments. P.K., A.V., and N.A.K. synthesized Cyst/Cd nanoparticles. J.S and R.H. wrote the manuscript. All authors reviewed and commented on the manuscript.

## Competing interests

The authors declare no competing interests.
