## [Peer Review File · Nature Communications]

Real-Time 3D Analysis During Tomographic Experiments using tomvizREVIEWER COMMENTS

Reviewer #1 (Remarks to the Author):

This paper outlines the capabilities of a live tomography module for tomviz, a tomographic reconstruction software. This new addition to the existing tomographic reconstruction and visualization software enables live reconstruction during experiment, or live visualization during reconstruction. The authors give an example of the utility of early structural assessment in handedness determination of a chiral object. They also point out advantages of viewing reconstruction progress as a reconstruction proceeds.

This manuscript clearly describes the use cases and features of the live tomography software. Electron tomography is widely used in many fields, and there are many advantages to performing reconstruction at the time of experiment as this software enables. As pointed out by the authors, many recent high impact scientific studies have relied on electron tomography, however these methods are known to require laborious analysis. This software will therefore likely be of interest to a broad user base performing impactful research. The reported software is novel and while other publications have reported live tomography [e.g. ref 9 of the manuscript], this is the only open source software able to perform live tomography that this reviewer is aware of.

In terms of the review criteria, the work is original, has a high probability to be impactful, and the claims of the manuscript regarding the software functionality are supported by the figures, supplementary videos, and my own testing of the software. As a report of a software for scientific studies, I would recommend publication of this manuscript in Nature Communications. I have some suggested changes that would improve the impact of the manuscript:

1. It would be interesting to hear the author's strategy for deploying live tomographic reconstruction on a high-performance computing cluster. Are there additional considerations a user should have in this scenario vs. on a personal computer?
2. Electron dose and subsequent beam damage are major limiting factors in electron tomography. It seems that live tomography enables structural insight at lower doses, as illustrated in Fig. 1. Comments on employing live tomography as a dose-minimizing strategy would likely be of interest to the readers of the manuscript.
3. The utility of live tomographic reconstruction is well illustrated for early classification of chiral structures. The utility of live viewing a reconstruction's progress after data collection is less clear, other than the ability to abort early if the reconstruction is not converging. More quantitative examples of this application would strengthen the manuscript.

Reviewer #2 (Remarks to the Author):

In this manuscript, J. Schwartz et al., demonstrated a real-time feedback capability during electron tomography in ADF-STEM settings, which are integrated into a software named "tomviz". Real-time feedback during the experiment is indeed important to fully utilize the microscope beamtime and achieve high-throughput 3D structural analysis. I think this can potentially be of interest to broad research communities which utilize high-resolution 3D structural analysis via electron tomography. However, I have several questions and issues regarding the manuscript that should be addressed before publication.

1. There are two main developments highlighted in the manuscript: 1) real-time feedback about the 3D structure during tomography experiments (the title of this manuscript focuses on this aspect), 2) real-time 3D visualization during tomographic reconstruction. While I fully appreciate the importance of the first part (real-time feedback during experiments), I am unsure about the usefulness of the second aspect. Usually, it requires enough iterations for an iterative

reconstruction algorithm to produce reasonable 3D tomograms. I doubt if there is any meaning to analyze the tomograms which have not converged yet. If the quality of the unconverged tomogram is not good, it can be due to improper reconstruction parameters, but it can also be due to not enough iterations. It would be more reasonable to tune the reconstruction parameters based on fully converged reconstructions to eliminate possible effects due to the iteration number. The authors should provide solid examples to show what importance the "live visualization of unconverged reconstructions" capability has.

2. It is possible to run tomography reconstruction during the experiment without this development; for example, one can run any tomographic reconstruction algorithm using the support computer after partial collection of the tilt series and check the result. The superb 3D fast visualization capability of tomviz is already demonstrated and published previously (Ref. 25 of the manuscript), so the new development here is about the real-time tomography capability. In this sense, it would be much more convincing if the authors can quantitatively show that their new development (the real-time capability) performs much better (it can be in terms of feedback speed, reconstruction quality, etc) than manually running the reconstruction using partial tilt series. They claim that volumetric interpretation can begin in less than 10 minutes and a high-quality tomogram is available within 30 minutes. Will it take much longer to get the volume if one runs manual reconstructions using the same number of tilt series images?

3. It would be great if the main idea which made the real-time tomography possible is clearly laid out in the manuscript. For example, the experimental section "Real-time Reconstruction Algorithms during Experimental Acquisition" contains some information, but it is not easy to see what exactly is the advancement. Is "the rescaling of the descent parameter" part really critical in implementing the real-time feedback during the experiment? Can't it be simply achieved by just running the SIRT algorithm with the additional experimental image using the previous reconstruction (reconstruction obtained without the newly obtained image) as an initial startpoint? It would be great if some more details about the meaning and importance of the new real-time reconstruction algorithm in the method section as well as in the main text.

4. The authors stated that "Here we present facile 3D visualization of specimens during an electron or cryo-electron tomography experiment using the tomviz platform (tomviz.org)." However, they demonstrated only one example of a real-time tomography experiment using ADF-STEM settings. There is no evidence about if this method would work for bright-field TEM experiments or cryoEM.

4.1. For high-resolution BF-TEM, proper CTF correction is necessary. Can the software handle the CTF correction in real-time during the experiment?

4.2. Even for low-resolution BF-TEM, the contrast should be properly inverted and processed to prepare the tilt series images to be linear projections of an object (requirement of tomography). Does the software take care of this in real-time during the experiment?

4.3. Accurate alignment of tilt series images to a center of rotation is a very essential step to obtain proper reconstruction, and to make the proper decision based on the real-time feedback during the experiment, it is important to have well-aligned tilt series. The authors have used the center-of-mass based approach, but this approach is very sensitive to background subtraction and only works for isolated objects. CryoEM tomography often utilizes the fiducial marker-based alignment, because the simple center-of-mass approach is not accurate enough. Does the simple center-of-mass based approach work for BF-TEM or cryoEM images for real-time experimental feedback? Can the software deal with the fiducial marker-based alignment during the real-time experiment?

To claim that the developed method generally works for TEM and cryoEM, the raised questions above (4.1 to 4.4) should be properly answered, and the authors should demonstrate at least one example for a real-time cryoEM tomography experiment. Otherwise, the authors should tone down their claim and just state that their new method is for ADF-STEM-based electron tomography.

5. As mentioned above, the center-of-mass based tilt series alignment can be very sensitive to background subtraction. The authors should provide enough details about how the background subtraction is performed in their software. Can it deal with a non-uniform background such as

carbon support?

6. Can the developed method (real-time feedback during the experiment) work for a non-isolated object? If it can, it would be great if an example can be demonstrated (at least in proper simulation which contains noise and spatial misalignment off from the center of rotation).

7. For the FePt data (reference 19 of the manuscript, physics.ucla.edu/research/imaging/FePt), it seems that they provide the raw tilt series and angles. However, the authors only have used the pre-processed data to demonstrate the live visualization during tomography reconstruction. It would be great if the authors can show that their real-time tomography experiment capability works with the raw unprocessed dataset and provides the initial reconstruction of reasonable quality required for making real-time decisions during the experiment.

Robert Hovden
Dept. Materials Science
University of Michigan
2700 Hayward Ave
Ann Arbor, MI 48109
Phone: (770) 265-4042

We thank the reviewers for their careful feedback and support for our project entitled “Real-Time 3D Analysis During Tomographic Experiments using tomviz” (Manuscript ID NCOMMS-21-45680A-Z). We appreciate the additional time required to time to review the manuscript, supplemental materials, and the software tool.

In addition to manuscript revisions, we have added new features, conducted experiments, and added three supplemental figures to meet the requests of both reviewers: **1.** We conducted a real-time TEM tomography experiment under cryogenic conditions (S. Fig 1 and S. Video 6). **2.** We added cross-correlation alignment as an optional pre-processing tool in tomviz and demonstrated its effectiveness on slab geometry specimens (S. Fig 3). **3.** We added CTF correction as an optional pre-processing tool in tomviz. **4.** We reconstruct the FePt dataset from raw data using our real-time pre-processing and reconstruction algorithms (now also serving as a replication study to Y. Yang *et. al.*).

This is the first report of real-time tomography (our non-peer reviewed conference presentations were made previously during its development) and will coincide with the tomviz 2.0 release containing features requested from this review.

Our responses are given in a point-by-point manner below.

Reviewer 1:

This paper outlines the capabilities of a live tomography module for tomviz, a tomographic reconstruction software. This new addition to the existing tomographic reconstruction and visualization software enables live reconstruction during experiment, or live visualization during reconstruction. The authors give an example of the utility of early structural assessment in handedness determination of a chiral object. They also point out advantages of viewing reconstruction progress as a reconstruction proceeds.

This manuscript clearly describes the use cases and features of live tomography software. Electron tomography is widely used in many fields, and there are many advantages to reconstruction at the time of experiment as this software enables. As pointed out by the authors, many recent high impact scientific studies have relied on electron tomography, however these methods are known to require laborious analysis. This software will therefore likely be of interest to a broad user base performing impactful research. The reported software is novel and while other publications have reported live tomography [e.g. ref 9 of the manuscript], this is the only open source software able to perform live tomography that this reviewer is aware of.

In terms of the review criteria, the work is original, has a high probability to be impactful, and the claims of the manuscript regarding the software functionality are supported by the figures, supplementary videos, and my own testing of the software. As a report of a software for scientific studies, I would recommend publication of this manuscript in Nature Communications. I have some suggested changes that would improve the impact of the manuscript:

We thank the reviewer for the positive comments and detailed suggestions on how to strengthen the manuscript (see below).

1. *It would be interesting to hear the author's strategy for deploying live tomographic reconstruction on a high-performance computing cluster. Are there additional considerations a user should have in this scenario vs. on a personal computer?*

Through code optimization and dynamically updating the reconstruction as data is collected we achieve speeds using a personal computer that maintain pace with typical experimental acquisitions (e.g. $512^3 - 1024^3$ voxels).

In this latest release, tomviz is implemented with an Intel thread building blocks (TBB) framework that allows multi-threaded processes for real-time visualization and is compatible with message passing interface (MPI) for multi-nodal parallelism required for high-performance computing. In this regard, we believe tomviz is now suited to serve future efforts in developing user-friendly, high-performance computing for real-time tomographic reconstruction of very large datasets—demand in this area is expected for beam-line x-ray CT.

On the user-manual on page 10 para. 2, we now illustrate how a user can most immediately make use of tomviz’s real-time visualization while processes run on a high-performance computer: “It is worth highlighting, tomviz can also monitor local or remote files in directories to dynamically update a 3D volume as the data changes. This means a reconstruction or volumetric process that is running remotely (e.g. on a high-performance cluster) can incrementally write the results to file and these updates will be rendered in tomviz. This approach readily accommodates computing (local or remote) that is ran outside of tomviz while simultaneously providing real-time 3D visualization.”

2. *Electron dose and subsequent beam damage are major limiting factors in electron tomography. It seems that live tomography enables structural insight at lower doses, as illustrated in Fig. 1. Comments on employing live tomography as a dose-minimizing strategy would likely be of interest to the readers of the manuscript.*

We agree that live tomography may provide an opportunity to reduce dose for some experiments. We have added a comment on page 4, para. 2 stating, “Although dose is fundamentally set by the experimental acquisition parameters (e.g. dwell time, beam current, sampling rate, tilt increment), in practice real-time tomography may reduce dose by streamlining acquisition and allowing early termination if reconstruction requirements are met.”

3. *The utility of live tomographic reconstruction is well illustrated for early classification of chiral structures. The utility of live viewing a reconstruction’s progress after data collection is less clear; other than the ability to abort early if the reconstruction is not converging. More quantitative examples of this application would strengthen the manuscript.*

We agree with the reviewer that ‘real-time 3D visualization during tomographic reconstruction’ has salient importance and utility. The second claim of this work “real-time 3D visualization” is also valuable for several reasons:

Developing the underlying threaded architecture that allows 3D data computation (or topographic reconstruction) simultaneous with volumetric visualization was pre-requisite to enabling experimental real-time tomography.

On page 3, para 1, we now expand our discussion with additional examples of how live visualization is valuable. “...Watching the convergence provides visual inspection and intuition to how hyperparameters influence the final 3D structure and ensures proper

convergence. For example, compressed-sensing inspired reconstruction methods are sensitive to regularization weights and require visual inspection to assess accuracy [1]. Furthermore, these advanced reconstruction algorithms do not exhibit predictable or monotonic convergence a priori and require monitoring to optimize convergence and determine when to terminate[2,3]. Even for traditional algorithms where convergence is more predictable, they are often slow and changes become marginal---the scientist need not wait to begin interpreting the 3D structure. Lastly, practical issues such as misalignment, spurious values in data (e.g. hot pixels), in-plane rotations, and other pre-processing artifacts alter or degrade a reconstruction, however, these problems are diagnosable without completing a full reconstruction. Real-time assessment saves researchers time by providing early feedback and optimizing reconstruction parameters to serve the longstanding goal of high-throughput tomography.”

[1] Jiang, Y. *et. al.* Sampling limits for electron tomography with sparsity-exploiting reconstructions *Ultramicroscopy* **186**, 94-103 (2018).

[2] Elfving, T., Nikazad, T., Hansen, P. Semi-convergence and relaxation parameters for a class of SIRT algorithms. *Electronic Transactions on Numerical Analysis* **37**, 321–336 (2010).

[3] Elfving, T., Hansen, P. C. & Nikazad, T. Semi-convergence properties of Kaczmarz’s method. *Inverse Problems* **30**, 055007 (2014).

Reviewer 2:

In this manuscript, J. Schwartz et. al. demonstrated a real-time feedback capability during electron tomography in ADF-STEM settings which are integrated into a software named “tomviz.” Real-time feedback during the experiment is indeed important to fully utilize the microscope beamtime and achieve high-throughput 3D structural analysis. I think this can potentially be of interest to broad research communities which utilize high-resolution 3D structural analysis via electron tomography. However I have several questions and issues regarding the manuscript that should be addressed before publication.

We thank the reviewer for the comments to strengthen the manuscript (see below).

1. *There are two main developments highlighted in the manuscript: 1) real-time feedback about the 3D structure during tomography experiments, 2) real-time 3D visualization during tomographic reconstruction. While I fully appreciate the importance of the first part, I am unsure about the usefulness of the second aspect. Usually it requires enough iterations for an iterative reconstruction to produce reasonable tomograms. I doubt if there is any meaning to analyze the tomograms which have not converged yet. If the quality of the unconverged tomograms is not good, it can be due to improper reconstruction parameters, but it can also be due to enough iterations. It would be more reasonable to tune the reconstruction parameters based on the fully converged reconstructions to eliminate possible effects due to the iteration number. The author should provide solid examples to show what importance the “live visualization of unconverged reconstructions” capability has.*

We agree with the reviewer that ‘real-time 3D visualization during tomographic reconstruction’ has salient importance and utility. The second claim of this work “real-time 3D visualization” is also valuable for several reasons:

Developing the underlying threaded architecture that allows 3D data computation (or topographic reconstruction) simultaneous with volumetric visualization is pre-requisite to enabling experimental real-time tomography.

Also, on page 3, para 1, we now expand our discussion with additional examples of how live visualization is valuable. “...Watching the convergence provides visual inspection and intuition to how hyperparameters influence the final 3D structure and ensures proper convergence. For example, compressed-sensing inspired reconstruction methods are sensitive to regularization weights and require visual inspection to assess accuracy [1]. Furthermore, these advanced reconstruction algorithms do not exhibit predictable or monotonic convergence a priori and require monitoring to optimize convergence and determine when to terminate[2,3]. Even for traditional algorithms where convergence is more predictable, they are often slow and changes become marginal---the scientist need not wait to begin interpreting the 3D structure. Lastly, practical issues such as misalignment, spurious values in data (e.g. hot pixels), in-plane rotations, and other pre-processing artifacts alter or degrade a reconstruction, however, these problems are diagnosable without completing a full reconstruction. Real-time assessment saves

researchers time by providing early feedback and optimizing reconstruction parameters to serve the longstanding goal of high-throughput tomography.” References [1-3] were added in this discussion.

[1] Jiang, Y. *et. al.* Sampling limits for electron tomography with sparsity-exploiting reconstructions *Ultramicroscopy* **186**, 94-103 (2018).

[2] Elfving, T., Nikazad, T., Hansen, P. Semi-convergence and relaxation parameters for a class of SIRT algorithms. *Electronic Transactions on Numerical Analysis* **37**, 321–336 (2010).

[3] Elfving, T., Hansen, P. C. & Nikazad, T. Semi-convergence properties of Kaczmarz’s method. *Inverse Problems* **30**, 055007 (2014).

*This response is provided in duplicate to reviewer #1 who had a similar request.

2. *It is possible to run tomography reconstruction during the experiment without this development; for example, one can run any tomographic reconstruction algorithm using the support computer after partial collection of the tilt series and check the result. The superb 3D fast visualization capability of tomviz is already demonstrated and published previously (Ref. 25 of the manuscript) so the new development here is about the real-time tomography capability. In this sense, it would be much more convincing if the authors can quantitatively show that their new development (the real-time capability) performs much better (it can be in terms of feedback speed, reconstruction quality, etc.) than manually running the reconstruction using partial tilt series.*

This article is the first peer-reviewed report of the ‘superb 3D fast visualization capability of tomviz’. Although our work was recently presented at the Microscopy and Microanalysis conference this work has not been published in a peer reviewed journal – the non-peer reviewed abstract (Ref. 25) was referenced at the editor’s request during submission. We apologize for the confusion. User feedback and adoption is essential for the success of our open-source platform. Without access to advertising budgets, we rely heavily on conference presentations and community engagement / adoption for testing and feedback.

They claim that volumetric interpretation can begin in less than 10 minutes and a high quality tomogram is available within 30 minutes. Will it take much longer to get the volume if one runs manual reconstructions using the same number of tilt series images?

Yes, we confirm that it will take longer to get the volume if one runs manual reconstructions using the same number of tilt series images. The real-time algorithms we implement do not re-run after new data is collected, but rather computation proceeds continuously during collection without restarting. By running the inverse problem continuously as new data arrives, missing information is filled-in in real time to maintain pace with experimental acquisition and completing roughly 50% faster than a reconstruction that begins after all data is collected (Ref. 20).

3. *It would be great if the main idea which made the real-time tomography possible is clearly laid out in the manuscript. For example, the experimental section “Real-time reconstruction algorithms during experimental acquisition” contains some information but it is not easy to see what exactly is the advancement.*

This manuscript presents a comprehensive software instrument that enables real-time electron tomography for the first time. In the discussion (page 4, para 2) we now state, “We achieved real-time electron tomography by integrating dynamic volumetric data analysis tools, data input / output, processing, reconstruction and visualization into a single software tool.” To illustrate tomviz’s substantiveness as a tool for science, on page 4 methods, we now include: “The entire package is built from 74,029 lines of code and 5,253 merges to date ... The entire code-base and dependencies are open source and compiled to maximize reproducibility.” Extensive testing, validation, and documentation (this manuscript) is also accompanied with this real-time electron tomography tool.

On page 1, para. 2, we now state “Iterative reconstruction algorithms efficiently accommodate new data and keep pace with typical experimental acquisition rates.”

Is the rescaling of the descent parameter part really critical in implementing the real-time feedback during the experiment? Can’t it be simply achieved by just running the SIRT algorithm with the additional experimental image using the previous reconstruction (reconstruction obtained without the newly obtained image) as an initial start point? It would be great if some more details about the meaning and importance of the new real-time reconstruction algorithm in the method section as well as in the main text.

SIRT benefits from a dynamic descent parameter to ensure that the process efficiently converges at a pace congruent with the experimental acquisition. We estimate the descent parameter through dynamic calculation of the Lipschitz constant, which also provides a simple and robust user experience.

On page 1, para 3 we now state, “Real-time algorithms accommodate the arrival of new data without restarting the reconstruction process. Iterative reconstruction methods are made efficient for real-time processes by utilizing dynamic descent parameters (See Methods). Dynamic reconstructions maintain pace with typical experimental acquisitions (e.g. $512^3 - 1024^3$ voxels) using a personal computer.”

4. *The authors stated that “Here we present facile 3D visualization of specimens during an electron or cryo-electron tomography experiment using the tomviz platform (tomviz.org).” However, they demonstrated only one example of a real-time tomography experiment using ADF-STEM settings. There is no evidence about if this method would work for bright-field TEM or cryoEM.*

We have now conducted cryogenic bright-field TEM tomography in real-time using the tomviz platform to directly demonstrate our reconstruction methods work for bright-field TEM and under cryogenic temperatures. Now Supplementary Figure S1 and Video 7 demonstrates a real-time tomography experiment of a nanoparticle cooled to 95 K (the nanoparticle is comprised of chiral dipeptide Cystine amino-acids).

4.1 For high-resolution BF-TEM, proper CTF correction is necessary. Can the software handle the CTF correction in real-time during the experiment?

We have now added a basic CTF correction tool. On page 3, para. 1 in the user manual we now state, “For BF-TEM, contrast inversion and CTF correction is often applied. The data can be inverted by Invert Data in Data Transforms (Fig. 3a) and CTF correction can be accessed in the Tomography dropdown menu (Fig. 3b). After CTF of the instrument is specified [1-3], the image data will be reweighted in Fourier space [4,5].”

[1] Rohou, A. & Grigorieff, N. CTFFIND4: Fast and accurate defocus estimation from electron micrographs. *J. Struct. Biol.* **192**, 216-221 (2015).

[2] Bell, J., Chen, M., Baldwin, P. & Ludtke, S. High Resolution single particle refinement in EMAN2.1. *Methods* **100**, 25-34 (2016).

[3] Tegunov, D. & Cramer, P. Real-time cryo-electron microscopy data preprocessing with Warp. *Nature Methods* **16**, 1146-1152 (2019).

[4] Minder, J.A. & Grigorieff, N. Accurate determination of local defocus and specimen tilt in electron microscopy. *J. Struct. Biol.* **142**, 334-347 (2003).

[5] Downing, K.H. & Glaeser, R.M. Restoration of weak phase-contrast images recorded with a high degree of focus: the twin image problem associated with CTF correction. *Ultramicroscopy* **108**, 921-928 (2008).

4.2 *Even for low-resolution BF-TEM, the contrast should be properly inverted and processed to prepare the tilt series images to be linear projections of an object (requirement of tomography). Does the software take care of this in real-time during the experiment?*

Yes, tomviz can readily handle contrast inversion and we value the referee's suggestion to make these features apparent to the reader. We now describe this feature on page 3 para. 1 in the user manual. In addition, we have included the ability to enable contrast inversion for real-time tomography reconstructions as well (Fig. 10 on page 7 for the user manual). Note also, because our platform uniquely allows for interactive adjustment of visualization parameters, altering the contrast opacity curve allows users to invert 3D visualization graphically in real-time as a reconstruction proceeds.

4.3 *Accurate alignment of tilt series images to a center of rotation is a very essential step to obtain proper reconstruction, and to make the proper decision based on the real-time feedback during the experiment. It is important to have a well-aligned tilt series. The authors have used the center-of-mass based approach but this approach is very sensitive to background subtraction and only works for isolated objects. CryoEM tomography often utilizes the fiducial marker-based alignment, because the simple center-of-mass approach is not accurate enough. Does the simple center-of-mass based approach work for BF-TEM or cryoEM images for real-time experimental feedback? Can the software deal with the fiducial marker-based alignment during the real-time experiment?*

We agree alignment is important. To address challenges associated with non-isolated objects we now incorporate cross correlation as an additional alignment method into the real-time tomography framework—now mentioned on page 6 para. 1 “We provide cross-correlation [1] for aligning non-isolated objects (Supplementary Fig S3).”

We have added Supplementary Figure S3 which compares center of mass with the added cross correlation on a specimen for a non-isolated object.

We now include references [2,3] to inform the reader on valuable work in fiducial marker-based alignment methods. Page 6 para. 3 we state, “For those that would like to perform marker-based alignment, we suggest using IMOD [2] prior to loading input projections in tomviz.”

[1] Frank, J. & McEwen, B. Alignment by cross-correlation. *Electron tomography* 205-213 (1992).

[2] Tegunov, D. & Cramer, P. Real-time cryo-electron microscopy data preprocessing with Warp. *Nature Methods* **16**, 1146-1152 (2019).

[3] Kremer, J., Mastrorade, D., & McIntosh, J. Computer visualization of three-dimensional image data using IMOD. *J. Struct. Biol.* **116**, 71-76 (1996).

S3 Comparison of center of mass alignment and cross correlation for non-isolated objects

Fig. S3 a) The phantom test object consisting two layers of nanoparticles inspired by an experimental Fe_3O_4 material system. b) The sinogram that corrupted by Poisson noise and misaligned by random translational image shifts. c-d) The sinograms aligned by center of mass and cross correlation and its resulting reconstructions. Scale bar, 10 nm.

To claim that the developed method generally works for TEM and cryoEM, the raised questions above (4.1 to 4.4) should be properly answered, and the authors should demonstrate at least one example for a real-time cryoEM tomography experiment. Otherwise, the authors should tone down their claim and just state their new method is for ADF-STEM based electron tomography.

We have conducted an additional experiment to demonstrate our real-time tomography tool works for TEM and cryogenic TEM tomography where SNR is not too low. Supplementary Figure S1 shows a real-time reconstruction experimentally collected at 95 K operating in TEM mode.

5. *As mentioned above, the center-of-mass based tilt series alignment can be very sensitive to background subtraction. The authors should provide enough details about how the background subtraction is performed in their software. Can it deal with non-uniform background such as*

carbon support?

We now provide additional details regarding the background subtraction in the methods section on page 6 para. 2 which states, “In our experiments, the uniform thin carbon support was removed by subtracting the average background scattered intensity.”

Generally, electron tomography benefits from a uniform background—specimens are prepared onto thin carbon films or silicon nitride membranes. However, some researchers may use lacey carbon supports when specimens are large or the cost of nanofabricated supports is prohibitive. Our tomographic reconstructions perform well on non-uniform lacey thin film supports to produce the full specimen and support structure.

6. *Can the developed method (real-time feedback during the experiment) work for a non-isolated object? If it can, it would be great if an example can be demonstrated (at least in proper simulation which contains noise and spatial misalignment off from the center of rotation).*

Yes, tomviz can work for non-isolated objects. We have added cross-correlation to the alignment process as shown in Supplementary Figure S3 with a synthetic dataset. By making our real-time approach modular and documented we have built a real-time platform that can be extended to scientific applications that require bespoke algorithms (e.g. Tutorial 5 in User Manual).

7. *For the FePt data (reference 19 of the manuscript) it seems that they provide the raw tilt series and angles. However the author only have used the pre-processed data to demonstrate the live visualization during tomography reconstruction. It would be great if the authors can show that their real-time tomography experiment capability works with the raw unprocessed dataset and provides the initial reconstruction of reasonable quality for making real-time decisions during the experiment.*

We want to re-iterate that our real-time tomography experiments have repeatably run on raw unprocessed data to “provide the initial reconstruction of reasonable quality required for making real-time decisions during the experiment.” (Supplementary Video 1)

However, here the referee is referring to the FePt data used in Figure 2. We agree with the reviewer that showing a real-time reconstruction of the unprocessed FePt data would strengthen the manuscript. In Supplemental Figure 2 we now show the atomic resolution reconstruction from raw, un-processed FePt data using our real-time compressed sensing algorithm. The FePt is a famous dataset demonstrating atomic resolution electron tomography [1] that has only been reconstructed (at least publicly) by one group of researchers using the GENFIRE algorithm [2].

By addressing the reviewers request, our work now serves as a replication study to atomic resolution electron tomography [1]. On page 3 para. 1, we briefly remark, “This work replicates Y. Yang’s atomic resolution tomogram using independent pre-processing and reconstruction methods”

[1] Yang, Y. *et. al.* Deciphering chemical order/disorder and material properties at the single-atom level. *Nature* **542**, 75-79 (2017).

[2] Pryor, A. *et. al.* GENFIRE: A generalized Fourier iterative reconstruction algorithm for high-resolution 3D imaging. *Sci. Rep.* **7**, 10409 (2017).

We feel that these changes have improved the manuscript and thank the referees for the helpful suggestions.

Please do not hesitate to contact us with any further comments or requests.

Sincerely yours,

Robert Hovden

REVIEWERS' COMMENTS

Reviewer #1 (Remarks to the Author):

The authors have added important information and clarifying statements to the manuscript which have fully addressed my comments. I recommend publication of this version of the manuscript.

Reviewer #2 (Remarks to the Author):

As I wrote in the original referee report, real-time feedback during the experiment is indeed important to fully utilize the microscope beamtime and achieve high-throughput 3D structural analysis. I think this can be of interest to broad research communities which utilize high-resolution 3D structural analysis via electron tomography.

I raised a number of concerns in my original report, and all of my comments, and those of the other referees, have been responded in a constructive way. Several unclear points in the manuscript have been well clarified, and the inclusion of cryogenic bright-field TEM tomography demonstration together with additional tilt-series preprocessing methods is convincing. I support publication of this manuscript in Nature Communications.